# Prevalence and Characterization of ESBL/AmpC Producing *Escherichia coli* from Fresh Meat in Portugal

**DOI:** 10.3390/antibiotics10111333

**Published:** 2021-11-01

**Authors:** Lurdes Clemente, Célia Leão, Laura Moura, Teresa Albuquerque, Ana Amaro

**Affiliations:** 1Laboratory of Bacteriology and Mycology, National Reference Laboratory of Animal Health, INIAV—National Institute of Agrarian and Veterinary Research, 2780-157 Oeiras, Portugal; celia.leao@iniav.pt (C.L.); lcroft.moura@gmail.com (L.M.); teresa.albuquerque@iniav.pt (T.A.); ana.amaro@iniav.pt (A.A.); 2CIISA—Centre for Interdisciplinary Research in Animal Health, Faculty of Veterinary Science, University of Lisbon, 1300-477 Lisbon, Portugal; 3MED—Mediterranean Institute for Agriculture, Environment and Development, 7006-554 Évora, Portugal; 4Faculty of Pharmacy Science, University of Lisbon, FFUL, 1649-019 Lisbon, Portugal

**Keywords:** antimicrobial resistance, *Escherichia coli*, meat, ESBL/AmpC, *mcr* genes, PMQR

## Abstract

The present study aimed to characterize the extended-spectrum β-lactamases and plasmid-mediated AmpC β-lactamases (ESBL/PMAβ) among *Escherichia coli* producers isolated from beef, pork, and poultry meat collected at retail, in Portugal. A total of 638 meat samples were collected and inoculated on selective medium for the search of *E. coli* resistant to 3rd generation cephalosporins. Isolates were characterized by antimicrobial susceptibility testing, molecular assays targeting ESBL/AmpC, plasmid-mediated quinolone resistance (PMQR), and plasmid-mediated colistin resistance (PMCR) encoding genes. The highest frequency of *E. coli* non-wild type to 3rd generation cephalosporins and fluoroquinolones was observed in broiler meat (30.3% and 93.3%, respectively). Overall, a diversity of acquired resistance mechanisms, were detected: *bla*_ESBL_ [*bla*_CTX-M-1_ (*n* = 19), *bla*_CTX-M-15_ (*n* = 4), *bla*_CTX-M-32_ (*n* = 12), *bla*_CTX-M-55_ (*n* = 8), *bla*_CTX-M-65_ (*n* = 4), *bla*_CTX-M-27_ (*n* = 2), *bla*_CTX-M-9_ (*n* = 1), *bla*_CTX-M-14_ (*n* = 11), *bla*_SHV-12_ (*n* = 27), *bla*_TEM-52_ (*n* = 1)], *bla*_PMAβ_ [*bla*_CMY-2_ (*n* = 8)], PMQR [*qnrB* (*n* = 27), *qnrS* (*n* = 21) and *aac(6’)-Ib-type* (*n* = 4)] and PMCR [*mcr-1* (*n* = 8)]. Our study highlights that consumers may be exposed through the food chain to multidrug-resistant *E. coli* carrying diverse plasmid-mediated antimicrobial resistance genes, posing a great hazard to food safety and a public health risk.

## 1. Introduction

Antimicrobial resistance in *Enterobacteriaceae* has increased in the past two decades, being extended-spectrum and plasmid-mediated AmpC β-lactamases (ESBL/PMAβ)-producing *Escherichia coli* a major concern for both human and animal health and food safety [1]. The use of antibiotics in humans and animals is probably the main factor for the increasing prevalence of ESBL/AmpC. However, other factors, particularly foreign travelling, international commercial trade of animals and products, animal movements, farming systems, animal husbandry, and the pyramidal structure of some types of primary production, likely also influence the spread of resistant clones [1,2]. Antibiotics other than third-generation cephalosporins, such as tetracyclines, sulfonamides and trimethoprim, widely used in animal therapy, may select ESBL/AmpC producers by co-selection; moreover, most encoding genes are plasmid-mediated [1].

Antimicrobial resistance (AMR) in food products stands at the interface of the animal and human sectors, mobile genetic elements playing an important role in its cross-sectorial transmission. Food can be contaminated with antimicrobial-resistant bacteria and antimicrobial resistance genes by the use of antibiotics during agricultural production, the possible presence of resistance genes in bacteria intentionally added to food products (starter cultures, probiotics), and through cross-contamination with antimicrobial resistant bacteria during food processing directly intended for human consumption [3,4].

ESBL and AmpC *Enterobacteriaceae* producers are widespread worldwide from samples of animal origin (food-producing, wild and companion animals), human, food products, and environmental (health care units, animal housing, slaughterhouses, surfaces, wastewater, retail stores) origins [5,6,7,8,9,10,11,12,13].

The potential for ESBL-producing *E. coli* strains from animals to cause human infections has long been suggested, yet not been confirmed or refuted [1,2,14]. Close genetic similarities between strains from broilers and their products and humans have been previously described [15,16,17].

Plasmid-mediated quinolone resistance (PMQR)-encoding genes have been commonly found in ESBL *Enterobacteriaceae* strains carried on plasmids, which can then be co-transferred to bacterial species of public health importance [8,18]. Although these determinants may be responsible for a low level of resistance, PMQR genes combined with chromosomal mechanisms are associated with a cumulative effect, explaining high minimum inhibitory concentration (MIC) values to nalidixic acid (≥128 mg/L) and ciprofloxacin (≥8 mg/L) [19]. In contrast, strains harboring PMQR genes tend to exhibit ciprofloxacin MIC values between 0.06 and 1 mg/L and nalidixic acid MIC values between 8 and 32 mg/L [20]. The occurrence of these genes may also increase the likelihood of dissemination of other resistance genes through co-selection [6,21,22]. PMQR determinants such as *qnrB, qnrS*, and *aac(6′)-lb-cr* seem quite common in Europe from different environments [20,23,24].

Following the first report of co-localization of plasmid-mediated colistin resistance (PMCR), *mcr-1* and ESBL [25], several studies described a worldwide distribution of *mcr-1* gene in ESBL-producing *Enterobacteriaceae* isolates from food and companion animals and meat [6,26,27,28,29]. Furthermore, studies have suggested that the use of extended-spectrum cephalosporins may have simultaneously favored the spread of *mcr-1* [25,26,27,28,29,30].

Although integrons are not considered mobile genetic elements per se, their location on plasmids and transposons enables gene transmission to inter and intra-species in a single event [31]. They can integrate, express, and disseminate gene cassettes carrying resistance determinants, playing a critical role in disseminating multidrug resistance (MDR) [32]. For this reason, they are increasingly reported worldwide, especially among *Enterobacteriaceae* [32].

The present study aims to evaluate the antimicrobial susceptibility, characterize the ESBL/PMAβ, and identify genes encoding fluoroquinolone and colistin resistance in *E. coli* isolates from retail meat (beef, pork and poultry) in Portugal.

## 2. Results

### 2.1. Bacterial Isolation

In this study, 109 non-wild type *E. coli* isolates resistant to third generation cephalosporins were isolated from 638 samples from beef (*n* = 26 out of 220), pork (*n* = 23 out of 220), and broiler meat (*n* = 60 out of 198). No isolates with reduced susceptibility to carbapenems were isolated (Table 1).

### 2.2. Antimicrobial Susceptibility Phenotypes

An overview of the antimicrobial susceptibility of 109 *E. coli* isolates is given in Table 2, where important parameters like MIC_50_ and MIC_90_ values, and rates of decreased susceptibility to critically and highly important antimicrobials for animals and humans are presented. Susceptibility profiles differed according to the origin of the isolates. As expected, all isolates showed decreased susceptibility to cefotaxime with MIC values ranging between 8 and 32 mg/L and slightly lower MIC values to ceftazidime (MIC= 4–8 mg/L).

Presumptive ESBL phenotype was prevalent and found in 87 isolates (79.8%), while presumptive AmpC phenotype was identified in 22 isolates (20.2%) (Figure 1).

Overall, the frequency of decreased susceptibility to fluoroquinolones was very high, particularly in poultry (93.3%). High levels of decreased susceptibility were also observed for tetracycline and trimethoprim in isolates from the three sources. Moderate levels of decreased susceptibility to azithromycin were found in beef and pork. Isolates from beef showed higher levels of decreased susceptibility to colistin (Table 2).

In this study, statistically significant differences (*p* ≤ 0.05) were observed between the three sources for the frequency of resistance to the following antibiotics: ciprofloxacin (*p* = 0.002), azithromycin (*p* = 0.002), and nalidixic acid (*p* = 0.002). Moreover, for MIC_50_ and MIC_90_ values, significant differences (difference of 3 to ≥8 dilutions) were observed for gentamicin (beef and poultry), chloramphenicol (beef and poultry), and ciprofloxacin (beef and pork) (Table 2).

Multidrug resistance was very high, particularly in isolates from poultry (95%), followed by pork (87%) and beef (84%) (Table 2). Overall, 44 multidrug resistance profiles were identified, of which decreased susceptibility to six and seven antibiotic groups was noted in all sources (Appendix A).

### 2.3. Molecular Characterization of Isolates

A high diversity of ESBL mechanisms was identified, being SHV-12 (27.6%) and CTX-M-1 (21.8%) the most prevalent, followed by CTX-M-32 (13.8%) and CTX-M-14 (12.6%). Of note, the identification of CTX-M-55 for the first time in Portugal in food products of animal origin (Table 3). Moreover, the rare occurrence of CTX-M-65, was registered in four isolates, (three from beef and one from pork) (Table 3). From the isolates with presumptive AmpC phenotype (*n* = 22, 20.2%), CMY-2 was found in eight isolates (7,4%) and an overexpression of the chromosomal AmpC gene was identified in fourteen (12.8%), mostly from poultry.

Among the isolates with reduced susceptibility to fluoroquinolones, *qnrB* and *qnrS* were the most commonly found PMQR mechanisms, found in 27 and 21 isolates, respectively; *aac(6′)-Ib-cr* was identified in four isolates (Table 3). Eight isolates from all sources showing decreased susceptibility to colistin harbored the *mcr-1* gene (Table 3).

Seventy-five isolates harbored class 1 (68.8%) or class 2 integrons (1.8%). In eight isolates (7.3%) both integrons class 1 and 2 were identified; no class 3 integrons were found in this study.

## 3. Discussion

In the present study, we aimed to establish the prevalence and the characterization of ESBL/AmpC *E. coli* producers in beef, pork, and poultry meat samples sold at retail. Our results show that 17.1% of all samples were contaminated, with the highest prevalence in poultry meat (30.3%), like that found in some European countries [33,34,35,36]. The prevalence in beef (11.8%) and pork (10.5%) samples was lower than in poultry, in accordance with that observed in Germany [8], suggesting that poultry meat might be easily more contaminated along the food chain than beef and pork [36]. Cross-contamination by contaminated flocks and the slaughterhouse environment greatly impacts the prevalence of ESBL/AmpC-producing *Enterobacteriaceae* in broiler chickens [36].

The use of third generation cephalosporins being interdict to poultry in the EU countries since 2009 [37], the higher occurrence of ESBL/AmpC *E. coli* producers might be assigned by the co-selection of other resistance mechanisms, by the use of various antibiotics, namely fluoroquinolones, polymyxins, tetracyclines, and sulfonamides, widely used in intensive animal production [38].

In this study, we observed a statistically significant difference for the frequency of resistance to ciprofloxacin (*p* = 0.002), azithromycin (*p* = 0.002), and nalidixic acid (*p* = 0.002) among the meat sources. Moreover, for MIC_50_ and MIC_90_ values, significant differences (3 to ≥8 dilutions) were observed for gentamicin (beef and poultry), chloramphenicol (beef and poultry), ciprofloxacin, and azithromycin (beef and pork), indicating the presence of two distinct microbial subpopulations, a susceptible and a resistant population [39]. The association between the frequency of resistance observed for the various antibiotics and the respective use in animals, among other factors, is in line with what has been described in other studies [40,41,42,43]. In food products, it is likely that a cross-contamination by people handling and preparing the meat, as well as on surfaces and with utensils in processing plants, may occur [44].

According to the European Food Safety Authority (EFSA), the prevalence of presumptive ESBL/AmpC *E. coli* producers in the different animal species and their products varies within the EU countries [45]. In the present study, eight types of ESBL encoding determinants from the CTX-M family were detected, indicating a high diversity of CTX-M-encoding genes in *E. coli* isolates from meat at retail in Portugal. The most common was CTX-M-1 identified in 19 isolates from beef, pork, and broiler meat, which is coherent with other studies [46,47,48,49,50,51]. CTX-M-32 and CTX-M-14 were also common and found in 12 and 11 isolates, respectively. CTXM-14 seems to be established in *Enterobacteriaceae* in European countries, including Portugal, as reported in studies over the last ten years performed in animals and food [5,6,9,18,52,53,54,55]. In one study from China in *E. coli* isolates from food, animals, and humans, CTX-M-14 was the most frequent ESBL found [56]. CTX-M-32 is an ESBL from CTX-M group 1 widely distributed in Portugal in *Enterobacteriaceae* isolates from animals, particularly pigs and food [5,8,9,57,58]. It has also been reported from some European and non-European countries [46,58,59], and in a study in the United Kingdom, CTX-M-32 was the most common variant found in dairy farms [53].

Being CTX-M-15, the most common ESBL found in human *Enterobacteriaceae* [60], was identified in five isolates from beef (*n* = 4) and pork (*n* = 1), seeming that broiler meat does not represent a source of *E. coli* CTX-M-15 producer, as described in the UK [50]. Of note, bovine is the animal group, compared to others, where CTX-M-15 has been recognized at a significant prevalence [61]. Thus, it may not be surprising to find CTX-M-15 producers in beef, suggesting that this colonization may rather originate from the animal sector [62].

The ESBL CTX-M-55 belongs to the CTX-M-1 group and is a derivative of CTX-M-15, differing by Ala80Val [63]. It is widely distributed in *E. coli* isolates from humans, animals, and meat in the Asian continent [56,64,65] and South America [66,67]. Although uncommon in Europe, it has been previously reported in humans, animals, and the environment [68,69,70,71]. In this study, it was detected in one isolate from beef and seven isolates from broiler, one being associated to SHV-12. In Portugal, previous studies [68] identified CTX-M-55 in clinical *E. coli* isolates from pets and humans. We note here its presence in food of animal origin.

The occurrence of CTX-M-65 in European countries is rare, although emerging. In this study, it was identified in four isolates, three from beef and one from pork. Genomic characterization by whole genome sequencing and genetic relatedness with other *E. coli* genomes suggest a clonal spread in Europe [72]. This enzyme is widely distributed in the Asian and North and South American continents [73,74,75,76], and only scarce studies reported its presence in Europe in humans, animals, and food [47,71,77,78,79].

CTX-M-27, a single nucleotide variant of CTX-M-14, seems to be emerging in certain parts of the world, namely in Asia and Europe [80]. This enzyme was the second most common ESBL found in human isolates in a hospital in Portugal [77], and was identified in healthy pigs at slaughter and in cats [6,67]. In this study, CTX-M-27 producers have higher MIC values to ceftazidime than those of CTX-M-14 producers, suggesting that the use of ceftazidime could be selecting CTX-M-27 [81].

CTX-M-9 was identified in one beef sample. In European countries, including Portugal, and from samples of animal origin, this enzyme seems to be more common in pigs, poultry, and poultry products [55,82,83,84], and rarer in cattle and beef [47]. In France, it was identified in *Enterobacteriaceae* producers from rescued wild birds [85].

SHV-12 was detected in 26 isolates, predominantly in chicken meat, either sole or associated with other β-lactamases of the CTX-M and TEM families. Similar results were obtained in other studies in food products [8,46,47,49,86].

In this study, TEM-52 was identified in one sample from broiler meat, seeming not to be as common as in the past, as the most frequent ESBL identified in retail poultry products in Portugal [55]. Likewise, a study in the Netherlands, found TEM-52 as the third ESBL identified in poultry meat samples, following CTX-M-1 and SHV-12 [47].

CMY-2 is the most common type of plasmid AmpC β-lactamase found in *Enterobacteriaceae* throughout the world, in isolates from human and animal origin. In this study, it was identified in eight isolates, which is in concordance with most research studies performed in the various countries, including Portugal [2,8,27,45,61,87,88,89].

In our study, 14 isolates (broiler meat, *n* = 13 and pork, *n* = 1) had an AmpC phenotype (cefoxitin MIC ≥ 32 mg/L) and MICs to cefotaxime lower (2–4 mg/L) than to ceftazidime (4–8 mg/L). Regarding cefepime, MIC was in the susceptible range, according to the clinical breakpoints (≤0.5 mg/L) established by EUCAST. As no plasmid-AmpC encoding determinants were detected, sequencing of the chromosomal *ampC* gene was performed, revealing highly conserved mutations in the promoter and attenuator region found in strong *ampC* promoters [90,91]. Indeed, these mutations have previously been described in *E. coli* isolates from food-producing animals and retail meat [92,93,94,95,96]. As hyperproduction of *ampC* gene is associated with clinically significant resistance to cephamycins and cephalosporins, decreasing the therapeutic options in the treatment of bacterial infectious diseases, its monitoring and detection are of the utmost importance [97,98,99].

Overall, the frequency of decreased susceptibility to colistin was not high (7.3%; 8/109). All colistin-resistant isolates (MIC > 2 mg/L) carried the *mcr-1* gene. Considering colistin one of the last-resort antibiotics to treat severe human infections caused by Gram-negative bacteria, and assuming that its consumption is probably the main factor for the increasing prevalence of resistant strains, the European Medical Agency (EMA) imposed a reduction of 50% on the sales of polymyxins in 25 of the 31 countries in the ESVAC network, for the period 2010–2018 [38]. Although Portugal has been decreasing its consumption since 2013, it is still the fifth country in Europe with higher consumption of polymyxins [38,100]. As colistin resistance is frequently associated with ESBL/AmpC producers, as in our study, selective pressure exerted by the consumption of third generation cephalosporins may contribute to co-selection of these resistance determinants [25].

Decreased susceptibility to fluoroquinolones was very high, particularly in isolates from poultry (93.3%), following the higher consumption of fluoroquinolones by this species, especially in Eastern and Southern European countries [34,38,100]. In 89 resistant isolates with a wide range of MICs (0.25 to ≥8 mg/L), PMQR were detected in 52, being *qnrB* the most frequent (*n* = 27), followed by *qnrS* (*n* = 21) and *aac(6′)-Ib-cr* (*n* = 4), in agreement with other studies [20]. Of note, the four CTX-M-65 producers co-harbored *qnrS* and *aac(6′)-Ib-cr* [72]. Studies in China in *Enterobacteriaceae* ESBL producers identified strains co-harboring two and three PMQR-encoding mechanisms simultaneously [76,101]. Although uncommon, in Europe, some *Enterobacteriaceae* strains from animals, food, and the environment co-harboring two PMQR encoding genes, were found, although not mentioning they were ESBL producers [20].

Horizontal transfer of antibiotic resistance genes among Gram-negative bacteria plays a major role in the spread of multidrug resistance, being integrons one of the most important drivers on its transmission. Their spread among microbial populations may be facilitated due to their location in transposons, such as *Tn*402 in class 1 and *Tn*7 in class 2 integrons, playing a relevant role as genetic reservoirs for transfer, integration, and dissemination of resistance genes among bacteria [102,103]. Previous studies underlined a higher prevalence of MDR in integron positive isolates [38,104]. In our study, 75 isolates carried class 1 integrons (68.8%) or class 2 (1.83%), which are in accordance with studies reporting that class 2 integrons are less prevalent in enteric bacteria [32].

Multidrug resistance was very high, particularly in isolates from poultry (95%), followed by pork (87%) and beef (84%), as described in other studies [8;27,51]; in 22 isolates, decreased susceptibility to six (*n* = 9) and seven (*n* = 13) antibiotic groups was noted. Forty-four MDR profiles were identified in this study (Appendix A). Beyond decreased susceptibility to β-lactams, the most common antibiotics shown in the various MDR patterns were quinolones, 79.5% (35/44), sulfamethoxazole, 84.1% (37/44), tetracycline, 75% (33/44), and trimethoprim, 70.5% (31/44), antibiotics widely used in intensive animal production [38]. The inappropriate and abusive use of antibiotics and the presence of mobile genetic elements promoting the spread of multiple resistance genes to other bacteria, as well as the circulation of specific multiresistant clones, may contribute to such a high prevalence of multidrug resistant strains [77,105].

Although plasmid-typing was not performed in this study, its role in the spread of several resistance traits is well-known, particularly some ESBL/AmpC-encoding genes carried in successful epidemic plasmids, being well adapted to different bacterial hosts, endowed with a very efficient conjugative system, or showing broad host range, or the capacity to be mobilized in trans by co-resident plasmids [106].

Being food products one of the sources of human exposure to antimicrobial-resistant microorganisms, the cross-transfer of ESBL/AmpC-encoding determinants is controversial, even though genomic investigations indicate that ESBL/AmpC encountered in non-human sources are distinct from those affecting humans, leading to the uncertainty on the real magnitude of their transfer to humans [14,107]. More recent studies speculate that continuous exposure to ESBL/AmpC *Escherichia coli* producers from non-human sources only sporadically result in a successful transfer of genes to human-adapted *E. coli* or other bacteria [108]. Of note, the risk to humans through food intake is also highly dependent on whether foodstuffs are consumed raw or cooked; nevertheless, cross-contamination may occur by people handling and preparing the meat, or by surfaces and kitchen utensils, if proper precautions are not taken.

## 4. Materials and Methods

### 4.1. Bacterial Isolation

Six hundred and thirty-eight samples from beef (*n* = 220), pork (*n* = 220), and broiler meat (*n* = 198) locally produced, were collected at random on the retail market throughout the country in 2016–2017, under the scope of monitoring and reporting of antimicrobial resistance in zoonotic and commensal bacteria (Commission Decision 652/2013) [109].

Laboratory procedures for the isolation and identification of extended-spectrum β-lactamase (ESBL/AmpC) and carbapenemase *Escherichia coli*-producers followed the protocols defined by the EURL-AR [110]. Briefly, 25 g of each meat sample was mixed with 225 mL of buffered peptone water (Oxoid, Basingstoke, UK), followed by incubation at 37 °C ± 1 °C for 18–22 h. Enriched samples were plated onto MacConkey Agar (Oxoid, UK) supplemented with 1 mg/L of cefotaxime (Glentham, Corsham, UK) and ChromID CarbaSmart (BioMérieux, France), followed by incubation at 44 °C ± 0.5 °C and 37 °C ± 1 °C for 18–22 h, respectively. Presumptive *E. coli* colonies were selected for biochemical identification on ColiID (bioMérieux, Marcy-l’Étoile, France) and those confirmed to be *E. coli* were subcultured and preserved at –80 °C for further tests.

### 4.2. Antimicrobial susceptibility testing

*E. coli* isolates from beef (*n* = 26), pork (*n* = 23), and broiler meat (*n* = 60) were tested for the determination of minimum inhibitory concentration (MIC), using the microdilution method and commercial standardized microplates (EUVSEC, Sensititre ™, Thermo Scientific™, Hampshire, UK). As all isolates were resistant toward cefotaxime, the second panel of antibiotics (EUVSEC2, Sensititre, Thermofisher Scientific, Hampshire, UK) was used to phenotypic detection of ESBL and AmpC enzymes. Results were interpreted according to the epidemiological breakpoint values of the European Committee on Antimicrobial Susceptibility Testing (EUCAST) [111]. *Escherichia coli* ATCC 25922 was used as a quality control strain.

In order to analyze results of susceptibility testing, values like MIC_50_ and MIC_90_ were calculated. The MIC_50_ represents the MIC value at which ≥50% of the isolates in a test population are inhibited, being equivalent to the median MIC value. The MIC_90_ represents the MIC value at which ≥90% of the strains within a test population are inhibited. Isolates were considered multidrug resistant (MDR) when resistance was evidenced to three or more antimicrobial classes [112].

### 4.3. Molecular Characterization of Antimicrobial Resistance Genes and Integrons

Total DNA was extracted using the boiling method [113]. DNA yield and purity check with a spectrophotometer (Nanodrop^®^ 2000, Thermo Scientific, Wilmington, NC, USA).

As all isolates evidenced a non-susceptible phenotype to third generation cephalosporins and or cephamycins, *bla*_ESBL_ (*bla*_TEM_, *bla*_SHV_, *bla*_OXA_, *bla*_CTX-M_) and *bla*_PMAβ_ (*bla*_CMY_, *bla*_MOX_, *bla*_FOX_, *bla*_LAT_, *bla*_ACT_, *bla*_MIR_, *bla*_DHA_, *bla*_MOR_, *bla*_ACC_)-encoding genes were screened by multiplex polymerase chain reaction (mPCR), as previously described [114]. The *E. coli* chromosomal *ampC* gene, including its promoter region, was also analyzed using Int-B2 and Int-H1 primers [99]. *Escherichia coli* ATCC 25922 was used as a standard control strain.

Additionally, isolates evidencing decreased susceptibility to fluoroquinolones and colistin were screened for the presence of PMQR (*qnrA*, *qnrB*, *qnrC*, *qnrD*, *qnrS*, *aac(6′)-Ib-cr*, *oqxAB*, and *qepA*) and plasmid-mediated colistin resistance genes (*mcr-1 to mcr-9*), using primers and conditions previously described [115,116,117]. The isolates were also subjected to the detection of class 1, 2, and 3 integrase encoding genes, as reported elsewhere [118,119]. Template DNAs from INIAV, German Federal Institute for Risk Assessment and the EURL-AR culture collections were used as positive controls in all PCR reactions.

PCR products of *bla* genes were purified with ExoSAP-IT™ (Applied Biosystems™, Fischer Scientific, Porto Salvo, Portugal), followed by Sanger sequencing using the BigDye^®^ Terminator v3.1 Cycle Sequencing Kit’s (Applied Biosystems™, Fischer Scientific, Porto Salvo, Portugal). Sequencing of fragments was performed in an automatic sequencer ABI3100 (Applied Biosystems, Warrington, UK). The identification of the *bla* gene variants was determined using the Basic Local Alignment Search Tool from the NCBI website [120] and The Comprehensive Antibiotic Resistance Database [121]. The chromosomal point mutations of the *ampC* gene were identified using BioEdit Sequence Alignment Editor version 7.2.5 [122] and BLAST with *E. coli* ATCC 25922 sequence as the reference genome.

### 4.4. Statistical Analysis

The Chi-square test was used to evaluate the statistical differences in the prevalence of antibiotic resistance between the matrices (beef, pork, and poultry). Fisher’s exact test was used as an alternative when the assumptions of the asymptotic analysis were not met. Statistical analysis was performed using the SPSS v26.0 (IBM). A probability value of *p* ≤ 0.05 was considered to indicate statistical significance.

## 5. Conclusions

In conclusion, this research paper complements and updates previous studies in Portugal, regarding antimicrobial resistance in *Enterobacteriaceae* from food products of animal origin. The investigation showed that *E. coli* could yield diverse plasmid-mediated antimicrobial resistance genes to critically important antimicrobials, posing a great hazard to food safety and a public health risk.

It also alerts for the emergence of ESBL variants uncommon in Europe like CTX-M-65 and CTX-M-55 in food products of animal origin. This study also reinforces previous reports that ESBL/AMPC producing *E. coli* has become one of the main indicators to estimate the burden of antimicrobial resistance in animals and other sectors from a One Health perspective.

## Figures and Tables

**Figure 1 antibiotics-10-01333-f001:**
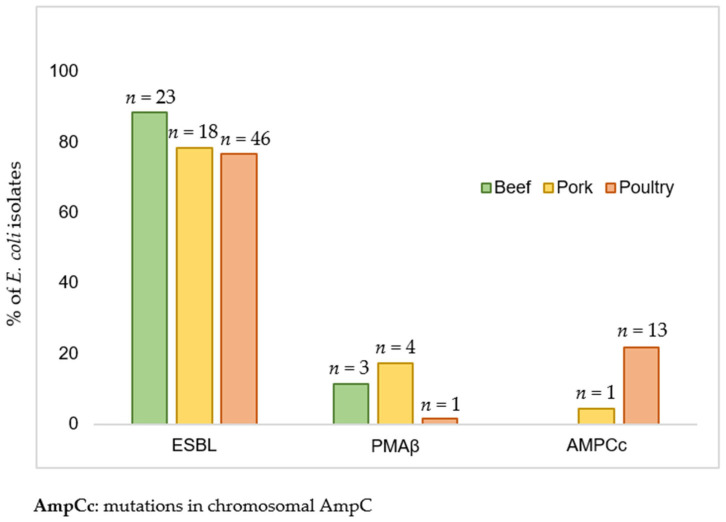
Prevalence of ESBL, PMAβ, and AmpCc phenotype in *Escherichia coli* from beef, pork, and poultry meat (*n* = 109).

**Table 1 antibiotics-10-01333-t001:** Number of samples collected and ESBL/AmpC *E. coli*-producers included in the study.

Animal Species/Source	N° of Samples Processed	N° of Samples Positive for *E. coli* CTX^R^ (%)
Beef	220	26 (11.8%)
Pork	220	23 (10.5%)
Poultry meat	198	60 (30.3%)
Total	638	109 (17.1%)

CTX^R^, cefotaxime-resistant *E. coli.*

**Table 2 antibiotics-10-01333-t002:** Antimicrobial susceptibility, MIC_50_, MIC_90_, and decreased susceptibility of 109 *Escherichia coli* isolates.

Antimicrobial	Beef (*n* = 26)	Pork (*n* = 23)	Poultry (*n* = 60)	ECOFFS ^c^
Ampicillin				
MIC_50_	>64	>64	>64	
MIC_90_	>64	>64	>64	
% DS	100	100	100	8
Cefotaxime				
MIC_50_	32	32	8	
MIC_90_	>64	>64	>64	
% DS	100	100	100	0.25
Ceftazidime				
MIC_50_	8	4	8	
MIC_90_	32	16	32	
% DS	96.2	95.7	96.7	0.5
Cefoxitin				
MIC_50_	8	8	4	
MIC_90_	32	64	64	
% DS	15.4	30.4	20	8
Cefepime ^b^				
MIC_50_	4	4	0.5	
MIC_90_	32	16	>32	
% DS	96,2	95.7	80	0.25
Nalidixic acid ^a^				
MIC_50_	>128	128	>128	
MIC_90_	>128	>128	>128	
% DS	69.2	56.5	91.7	8
Ciprofloxacin ^a,b^				
MIC_50_	0.5	0.5	8	
MIC_90_	>8	>8	>8	
% DS	69.2	65.2	93.3	0.064
Azitromycin ^a^				
MIC_50_	8	8	8	
MIC_90_	64	>64	16	
% DS	30.8	30.4	11.7	(8) ^c^
Gentamicin ^b^				
MIC_50_	1	≤0.5	1	
MIC_90_	32	2	>32	
% DS	19.2	8.7	15	2
Chloramphenicol ^b^				
MIC_50_	≤8	≤8	≤8	
MIC_90_	>128	64	128	
% DS	46.2	21.7	41.7	16
Colistin				
MIC_50_	≤1	≤1	≤1	
MIC_90_	4	≤1	≤1	
% DS	15.4	4.3	5	2
Tetracycline				
MIC_50_	64	>64	>64	
MIC_90_	>64	>64	>64	
% DS	73.1	82.6	86.7	8
Sulphamethoxazole				
MIC_50_	>1024	>1024	>1024	
MIC_90_	>1024	>1024	>1024	
% DS	ND	ND	ND	ND
Trimethoprim				
MIC_50_	>32	>32	>32	
MIC_90_	>32	>32	>32	2
% DS	76.9	56.5	68.3	
MDR (%)	84	87	95	

DS, decreased susceptibility; ND, not determined; MDR, multidrug resistant; ^a^ *p* ˂ 0.05; ^b^ ≥3 dilution steps between MIC_50_ and MIC_90_ values; ^c^ tentative epidemiological cut-off.

**Table 3 antibiotics-10-01333-t003:** Genotypic characterization of ESBL and AmpC producing-*Escherichia coli* (*n* = 109).

*bla* Genes (*n*)	PMCR *	PMQR **			Integrons		
	*mcr-1* (*n*)	*qnrB* (*n*)	*qnrS* (*n*)	*aac(6′)Ib-cr* (*n*)	Class 1 (*n*)	Class 2 (*n*)	Class 1 and 2 (*n*)
**Broiler meat (*n* = 60)**							
*bla*_CTX-M-1_ (*n* = 11)		6			5		2
*bla*_CTX-M-32_ (*n* = 3)		1			3		
*bla*_CTX-M-55_ (*n* = 6)	1	1	4		6		
*bla*_CTX-M-55_ + *bla*_SHV-12_ (*n* = 1)					1		
*bla*_CTX-M-14_ (*n* = 4)	2		1		4		
*bla*_TEM-52_ (*n* = 1)							
*bla*_SHV-12_ (*n* = 17)		3	2		13		
*bla*_SHV-12_ + *bla*_TEM-1_ (*n* = 3)		2			3		1
*bla*_CMY-2_ (*n* = 1)			1				
*bla*_AmpCc_ (*n* = 12)		4			10		1
*bla*_AmpCc_ + *bla*_TEM1_ (*n* = 1)		1			1		
**Beef (*n* = 26)**							
*bla*_CTX-M-1_ (*n* = 6)			1		2	1	
*bla*_CTX-M-15_ (*n* = 4)			2		3		
*bla*_CTX-M-55_ + *bla*_TEM-1_ (*n* = 1)					1		
*bla*_CTX-M-65_ (*n* = 2)	2		2	2	2		
*bla*_CTX-M-65_ + *bla*_TEM-1_ + *bla*_SHV-12_ (*n* = 1)			1	1	1		
*bla*_CTX-M-9_ (*n* = 1)					1		
*bla*_CTX-M-14_ (*n* = 4)	1	1	1		4		
*bla*_SHV-12_ + *bla*_TEM-1_ (*n* = 2)							1
*bla*_SHV-12_ (*n* = 2)	1	1			1		
*bla*_CMY-2_ (*n* = 2)							1
*bla*_CMY-2_ + *bla*_TEM-1_ (*n* = 1)					1		
**Pork (*n* = 23**)							
*bla*_CTX-M-1_ (*n* = 2)					1		1
*bla*_CTX-M-15_ (*n* = 1)		1			1		
*bla*_CTX-M-32_ (*n* = 9)		2	2		5	1	1
*bla*_CTX-M-65_ + *bla*_TEM-1_ (*n* = 1)	1		1	1	1		
*bla*_CTX-M-27_ (*n* = 2)			2				
*bla*_CTX-M-14_ (*n* = 3)		1			3		
*bla*_CMY-2_ (*n* = 3)		1					
*bla*_CMY-2_ + *bla*_TEM-1_ (*n* = 1)			1		1		
*bla*_AmpCc_ (*n* = 1)		1			1		

* Plasmid-mediated colistin resistance genes; ** plasmid-mediated quinolone resistance genes; AmpCc: chromosomal AmpC mutation.

## Data Availability

The data that support the findings of this study are available within the article.

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
