# Peer review of "Prevalence and Characterization of ESBL/AmpC Producing Escherichia coli from Fresh Meat in Portugal"

_antibiotics, 2021, doi:10.3390/antibiotics10111333_

Round 1

Reviewer 1 Report

The author’s covers important food safety area on the emergence and presence of extended-spectrum and plasmid-mediated AmpC (ESBL/PMAβ) β-lactamase Escherichia coli from food of animal sources in Portugal. Authors found 109 E. coli isolates from 638 meats samples. The authors found highest frequency of resistance in E. coli against 3rd generation cephalosporins and fluoroquinolones from broiler meat. Authors found many blaCTX-M-1 and CTX-M-55 which are the most abundant CTM-M types from animal sources in other literature. Some clinically relevant blaCTX-M-15, quinolones and colistin resistance determinants (mcr-1) were also observed in this study.

I have following comments.

  1. Why the authors select E. coli only, give rationale?
  2. The article title is bit misleading to Critically Important Antimicrobials should be removed as not all the CIA resistance mechanisms were studies.
  3. Title should be changed in the context as resistance mechanisms were not investigated in depth rather it is a descriptive study.
  4. The authors selected cefotaxime resistant coli only, and then looked co-resistance to other antibiotics among ESBL/AmpC producers. Therefore, this study and title should clarify these points.
  5. Further investigations into plasmid types and STs could have improved the spectrum of study.

Line 41-44: The statement is vague.

Line 107-108 Why resistance most of the other antibiotics are higher in beef than in pork and poultry isolates? In line 117, it is said most of antibiotics are resistance from broiler sources.

Section 2.2: Define what is MDR with reference?

Author Response

We thank all the reviewers for the time spent reviewing the manuscript.

Changes in the text are highlighted in yellow accordingly to the reviewer’s suggestions and comments.

REVIEWER 1

  1. As mentioned in line 108-112, the samples used in this study were collected under the scope of the surveillance program from the Commission Decision 652/2013 - “Monitoring and reporting of antimicrobial resistance in zoonotic and commensal bacteria -, which defines that only ESBL/AmpC and carbapenemases coli producers are searched in retail meat samples.

2, 3 and 4. The title was changed to accommodate the reviewer’s suggestion:

Prevalence and characterization of ESBL/AmpC producing Escherichia coli from fresh meat in Portugal

5. We agree with the reviewer stating that further investigations into plasmid-typing could have improved the spectrum of the study.

Line 41-44

    “ Antibiotics other than third-generation cephalosporins can select ESBL/AmpC producers, such as tetracyclines, sulfonamides and trimethoprim, widely used in animal therapy, as many ESBL/AmpC-encoding genes are mostly located on plasmids carrying other multidrug resistance determinants [1]”.

     Replace by another sentence:

     Antibiotics other than third-generation cephalosporins, such as tetracyclines, sulfonamides and trimethoprim, widely used in animal therapy, may select ESBL/AmpC producers by co-selection; moreover, most encoding genes are plasmid-mediated [1].

Line 107-108

“Overall, the frequency of decreased susceptibility to the other antimicrobials (azithromycin, colistin, gentamicin, chloramphenicol, tetracycline and trimethoprim) was higher in beef than in pork and poultry isolates, excepting for fluoroquinolones, which was higher in poultry (93.3%), compared with beef (69.2%) and pork (65.2%)

 Replace by another sentence:            

Overall, the frequency of decreased susceptibility to fluoroquinolones was very high, particularly in poultry (93.3%). High levels of decreased susceptibility were also observed for tetracycline and trimethoprim in isolates from the three sources. Moderate levels of decreased susceptibility to azithromycin were found in beef and pork. Isolates from beef showed higher levels of decreased susceptibility to colistin (Table 2).

Section 2.2.

The following sentence was introduced: “Isolates were considered multidrug resistant (MDR) when resistance was evidenced to three or more antimicrobial classes [113].”

Reviewer 2 Report

Lurdes et al presented a timely topic as antibiotic resistance has become a great threat to human health globally. The author characterised the extended-spectrum and plasmid-mediated AmpC (ESBL/PMAβ) β-lactamase Escherichia coli producers isolated from 638 meat samples in Portugal. E coli resistance phenotype has been well correlated with the molecular characterisation. I do not have further comments. The amount of the present work is impressive and the manuscript is well written. I don't have further comments.

Author Response

REVIEWER Nº2

We thank the reviewer Nº 2 for the kind comments to our study.

Reviewer 3 Report

This study updates previous studies in Portugal, regarding antimicrobial resistance in Enterobacteriaceae from food products of animal origin. The investigation showed that E. coli could carry diverse plasmid-mediated antimicrobial resistance genes to critically important antimicrobials. It also alerts for the emergence of ESBL variants uncommon in Europe like CTX-M-65 and CTX-M-55 in food products of animal origin. This study deals with a cutting edge theme, which is very interesting for the audience of this journal. It provides valuable results and encourages research to investigate diverse plasmid-mediated antimicrobial resistance genes to critically important antimicrobials, and posing a great hazard to food safety and a public health risk. Congratulations on the study and the results obtained.

Minor issues:

1) Line 16: “…the extended-spectrum and plasmid-mediated AmpC (ESBL/PMAβ) β-lactamase Escherichia coli …;” "… extended-spectrum β-lactamases and plasmid-mediated AmpC β-lactamases (ESBL/PMAβ) among Escherichia coli …", for me is better; check in the main text and correct.

2) Line 114: “(Schwarze et al., 2010)”, delete this reference from the results section.

3) Table 2: correct the ECOFFS value in correspondence with Azithromycin (81), and specify in full the abbreviation indicated, as indicated for the others under the table.

Author Response

REVIEWER Nº3

We thank the reviewer for the kind comments to our study. All the reviewer suggestions were accepted.

Line 16

Text changed and highlighted in yellow.

Line 114:

Reference withdrawn from the text

Table 2

Changes highlighted in yellow